# Learning Heuristics for Automated Reasoning through Reinforcement Learning

## Abstract

We demonstrate how to learn efficient heuristics for automated reasoning algorithms through deep reinforcement learning. We focus on backtracking search algorithms for quantified Boolean logics, which already can solve formulas of impressive size - up to 100s of thousands of variables. The main challenge is to find a representation of these formulas that lends itself to making predictions in a scalable way. For challenging problems, the heuristic learned through our approach reduces execution time by a factor of 10 compared to the existing handwritten heuristics.

## 1 Introduction

Automated reasoning and machine learning have both made tremendous progress over the last decades. Automated reasoning algorithms are now able to solve challenging logical problems that once seemed intractable, and today are used in industry for the verification of hardware and software systems. At the core of this progress are logic solvers for Boolean Satisfiability (SAT), Satisfiability Modulo Theories (SMT), and recently also Quantified Boolean Formulas (QBF). These solvers rely on advanced backtracking search algorithms, such as conflict-driven clause learning (CDCL) (Marques-Silva & Sakallah, 1997), to solve formulas with thousands or even millions of variables. Many automated reasoning algorithms are not only able to answer challenging queries, but also to explain their answer by providing detailed mathematical proofs. However, when problems lack a formal semantics automated reasoning algorithms are hard to apply, and they tend to scale poorly for problems involving uncertainty (e.g. in form of probabilities).

Machine learning, on the other hand, recently experienced several breakthroughs in object recognition, machine translation, board games, and language understanding. At the heart of the recent progress in machine learning lie optimization algorithms that train deep and overparameterized models on large amounts of data. The resulting models generalize to unseen inputs and can generally deal well with uncertainty. However, the predictions made by learned models are often hard to explain and hard to restrict to safe behaviors.

The complementary strengths of machine learning and automated reasoning raise the question how the methods of the two fields can be combined. In this paper we study how neural networks can be employed within automated reasoning algorithms. We discuss the unique challenges that arise in applying neural networks to vast logical formulas and how reinforcement learning can be used to learn better heuristics for backtracking search algorithms.

Backtracking search algorithms in automated reasoning are highly nondeterministic: at many points in their execution there are multiple options for how to proceed. These choices have a big impact on performance; in fact we often see that some problems cannot be solved within hours of computation that under different heuristics are solved within milliseconds. The most prominent heuristic choice in CDCL-style search algorithms is the *variable selection heuristic*, which is to select a variable (and which value to assign to that variable). Until today, Variable State Independent Decaying Sum (VSIDS) (Moskewicz et al., 2001b) and its variants are used as the variable selection heuristic in some of the most competitive SAT solvers (Biere, 2010; Soos, 2014; Audemard & Simon, 2014) and QBF solvers (Lonsing & Biere, 2010; Rabe & Seshia, 2016). Even small improvements over VSIDS, such as the recent introduction of learning rate branching (Liang et al., 2016), are highly interesting for the automated reasoning community.

Designing heuristics for backtracking search algorithms is often counter-intuitive: For example, it is well-known that heuristics that aim to find a correct solution perform worse than heuristics that aim to quickly cause a *conflict* (i.e. reach a dead-end in the search tree). This is because running into a conflict allows us to apply *clause learning*, a process that (deductively) derives a new constraint that prevents running into the similar conflicts again. This can cut off an exponential number of branches and can therefore save a lot of computation time in the future. Heuristics for search algorithms are an interesting target for machine learning with immediate impact in verification.

**The Problem**   We want to learn a heuristic for a backtracking search algorithm such that it solves more formulas in less time. We view this as a reinforcement learning problem where we are given a formula from some distribution of formulas, and we run the backtracking search algorithm until it reaches a decision point and then query the neural network to predict an action. We iterate this process until termination (at which point the formula is proven or refuted) or a time limit is reached. The input to the neural network is thus the current state of the algorithm, including the formula, and we reward it for reaching termination quickly.

In this work, we focus on learning heuristics for the solver CADET (Rabe & Seshia, 2016; Rabe et al., 2018). CADET is a competitive solver for QBF formulas with the quantifier prefix $\forall X \exists Y$, which covers most of the known applications of QBF. For all purposes of this work, we can assume the algorithm works exactly like a SAT solver. The heuristic we address in this work is the variable selection heuristic.

**Challenges**   The setting comes with several unique challenges:

*Size:*   While typical reinforcement learning settings have a small, fixed-size input (a Go board or an Atari screen), the formulas that we consider have hundreds of variables and thousands of clauses, each with several data points. All variables appear both as part of the state and as actions in the reinforcement learning environment. At the same time, the size of the formulas varies dramatically (from tens of variables up to 700k variables) and is hardly correlated with their 'difficulty' - while some of the largest formulas can be solved with little effort, some of the formulas with only 100 variables cannot be solved by any modern QBF solver.

*Lenth:*   The length of episodes (=run of the algorithm on a given formula) varies a lot and can easily exceed 100k steps. For many of the formulas, we have, in fact, never observed a terminating run.

*Performance:*   Our aim is to solve formulas in less time. While learning how to take clever decisions can lead us to the solution in less steps, the added cost of computing these decisions can outweigh the benefit. Typical inference times of neural networks are in the range of milliseconds, but this is 10x to 100x more than it takes the execution to reach the next decision point.[1] Hence, any deep learning approach has to to pick drastically better decisions in order to overcome this gap, while relying on relatively cheap models.

*Nameless variables:*   Variable $x$ in one formula shares basically no meaning with variable $x$ in another formula. Even worse, our variables are nameless and only assigned a number.

**Contributions**   We demonstrate that it is possible to create superior heuristics for automated reasoning algorithms using deep reinforcement learning. The approach we explore in this work builds on Graph Neural Networks (GNNs) (Scarselli et al., 2009) and overcomes the challenges of the size of formulas, length of the episodes, performance, and nameless variables.

GNNs allow us to compute an embedding of each variable based on their context *in the current formula*, instead of learning fixed embeddings for variables that are shared across all formulas. This solves the problem of relying on names, and allows us to scale the approach to formulas with arbitrary numbers of variables. We predict the quality of each variable as a decision variable based only on its embedding. This means that decisions are being taken only with local information. While in theory this limits the power of the heuristics we can learn, this approach comes with great generalization properties: We show that training a heuristic on small and easy formulas helps us to solve much larger and harder formulas from similar distributions.

After a primer on Boolean logics in Section 2 we define the problem in Section 3, and describe the network architecture in Section 4. We describe our experiments in Section 5.

---

[1] For SAT solvers, this factor one or two orders of magnitude larger than for the QBF solver we focus on, which is one of the reasons we focus on QBF in this work.

## 2 BOOLEAN LOGICS AND SEARCH ALGORITHMS

We start with describing *propositional* (i.e. quantifier-free) Boolean logic. Propositional Boolean logic allows us to use the constants 0 (false) and 1 (true), variables, and the standard Boolean operators $\land$ (*"and"*), $\lor$ (*"or"*), and $\neg$ (*"not"*).

A *literal* of variable $v$ is either the variable itself or its negation $\neg v$. By $\bar{l}$ we denote the logical negation of literal $l$. We call a disjunction of literals a *clause* and say that a formula is in *conjunctive normal form* (CNF), if it is a conjunction over clauses. For example, $(x \lor y) \land (\neg x \lor y)$ is in CNF. It is well known that any Boolean formula can be transformed into CNF. It is less well known that this increases the size only linearly, if we allow the transformation to introduce additional variables (Tseitin transformation (Tseitin, 1968)). Thus, we can assume that all formulas are given in CNF.

### 2.1 DPLL AND CDCL

The satisfiability problem of propositional Boolean logics (SAT) is to find a satisfying assignment for a given Boolean formula or to determine that there is no such assignment. SAT is the prototypical NP-complete problem and many other problems in NP can be easily reduced to it. The first backtracking search algorithms for SAT are attributed to Davis, Putnam, Logemann, and Loveland (DPLL) (Davis & Putnam, 1960; Davis et al., 1962). Backtracking search algorithms gradually extend a partial assignment until it becomes a satisfying assignment, or until a *conflict* is reached. A conflict is reached when the current partial assignment violates one of the clauses and hence cannot be completed to a satisfying assignment. In case of a conflict, the search has to backtrack and continue in a different part of the search tree.

*Conflict-driven clause learning* (CDCL) is a significant improvement over DPLL due to Marques-Silva and Sakallah (Marques-Silva & Sakallah, 1997). CDCL combines backtracking search with *clause learning*. While DPLL simply backtracks out of conflicts, CDCL "analyzes" the conflict by performing a couple of *resolution* steps. Resolution is an operation that takes two existing clauses $(l_1 \lor \cdots \lor l_n)$ and $(l'_1 \lor \cdots \lor l'_n)$ that contain a pair of complementary literals $l_1 = \neg l'_1$, and derives the clause $(l_2 \lor \cdots \lor l_n \lor l'_2 \lor \cdots \lor l'_n)$. The conflict analysis adds new clauses over time, but thereby helps to cut off large parts of the search space.

Since the introduction of CDCL in 1997, countless refinements of CDCL have been explored and clever data structures improved its efficiency significantly (Moskewicz et al., 2001a; Eén & Sörensson, 2003; Goldberg & Novikov, 2007). Today, the top-performing SAT solvers, such as Lingeling (Biere, 2010), Crypominisat (Soos, 2014), Glucose (Audemard & Simon, 2014), and MapleSAT (Liang et al., 2016), all rely on CDCL and they solve formulas with millions of variables for industrial applications such as bounded model checking (Biere et al., 2003).

### 2.2 QUANTIFIED BOOLEAN FORMULAS

QBF extends propositional Boolean logic by *quantifiers*, which are statements of the form "for all $x$" ($\forall x$) and "there is an $x$" ($\exists x$). The formula $\forall x.\ \varphi$ is true if, and only if, $\varphi$ is true if $x$ is replaced by 0 (false) and also if $x$ is replaced by 1 (true). The semantics of $\exists$ arises from $\exists x.\ \varphi = \neg \forall x.\ \neg \varphi$. We say that a QBF is in prenex normal form if all quantifiers are in the beginning of the formula. W.l.o.g., we will only consider QBF that are in prenex normal form and whose propositional part is in CNF. Further, we assume that for every variable in the formula there is exactly one quantifier in the prefix. An example QBF in prenex CNF is $\forall x.\ \exists y.\ (x \lor y) \land (\neg x \lor y)$.

We focus on 2QBF, a subset of QBF that admits only one quantifier alternation. W.l.o.g. we can assume that the quantifier prefix of formulas in 2QBF starts with a sequence of universal quantifiers $\forall x_1 \ldots \forall x_n$ followed by a sequence of existential quantifiers $\exists y_1 \ldots \exists y_m$. While 2QBF is less powerful than QBF, we can encode many interesting applications from verification and synthesis, e.g. program synthesis (Solar-Lezama et al., 2006; Alur et al., 2013). The algorithmic problem considered for QBF is to determine the truth of a quantified formula (TQBF). After the success of CDCL for SAT, CDCL-like algorithms have been explored for QBF as well (Giunchiglia et al., 2001; Lonsing & Biere, 2010; Rabe & Seshia, 2016; Rabe et al., 2018). We focus on CADET, a solver that implements a generalized CDCL backtracking search algorithm (Rabe & Seshia, 2016; Rabe et al., 2018). It is one of the state-of-the-art solvers for 2QBF and has the ability to prove its results, which allows us to ensure that the reinforcement learning algorithm does not find any loopholes in the environment.

## 3 PROBLEM DEFINITION

In this section, we first revisit reinforcement learning and explain how it maps to the setting of logic solvers. In reinforcement learning, we consider an agent that interacts with an environment $\mathcal{E}$ over discrete time steps. The environment is a Markov decision process (MDP) with states $\mathcal{S}$, action space $\mathcal{A}$, and rewards per time step denoted $r_t \in \mathbb{R}$. A *policy* is a mapping $\pi : \mathcal{S} \times \mathcal{A}$, such that $\sum_{a \in \mathcal{A}} \pi(s, a) = 1\ \forall s$, defining the probability to take action $a$ in state $s$. The goal of the agent is to maximize the expected (possibly discounted) reward; formally $J(\pi) = \mathbb{E}\left[\sum_{t=0}^{\infty} \gamma^t r_t | \pi \right]$.

In our setting, the environment $\mathcal{E}$ is the solver CADET (Rabe & Seshia, 2016). The environment is deterministic except for the initial state, where a formula is chosen randomly from a distribution. At each time step, the agent gets an observation, which consists of the QBF formula and the solver state. The agent has to select from a subset of the available variables (only variables that do not have a value yet are valid actions). Formally, the space of actions is the set of all variables in all possible formulas in all solver states, where at every state only a small finite number of them is available. Practically, the agent will never see the effect of even a small part of these actions, and so it must generalize over unseen actions, by learning dense embeddings of actions in $\mathbb{R}^n$. We assume that the set of available actions is part of the observation.

An *episode* is the result of the interaction of the agent with the environment. We consider an episode to be *complete*, if the solver reaches a terminating state in the last step. As there are arbitrarily long episodes, we want to abort them after some step limit and consider these episodes as *incomplete*.

### 3.1 BASELINES

While there are no competing learning approaches yet, human researchers and engineers have tried many heuristics for selecting the next variable. VSIDS is the best known heuristic for the solver we consider (and is still one of the state-of-the-art heuristics in SAT solving). We therefore consider VSIDS as the main baseline. VSIDS maintains an *activity score* per variable and always chooses the variable with the highest activity that is still available. The activity reflects how often a variable recently occurred in conflict analysis. To select a literal of the chosen variable, VSIDS uses the Jeroslow-Wang heuristic (Jeroslow & Wang, 1990), which selects the polarity of the variable that occurs more often, weighted by the size of clauses they occur in. For reference, we also consider the *Random* heuristic, which chooses one of the available actions uniformly at random.

## 4 THE NEURAL NETWORK ARCHITECTURE

Our model has to read a formula and the state of the algorithm, and select one of the available variables. The representational challenges to applying deep learning techniques in this setting include that the input can be very large, may have vastly different size depending on the formula, and the action space is infinite (though only a finite subset of the actions is available in any step).

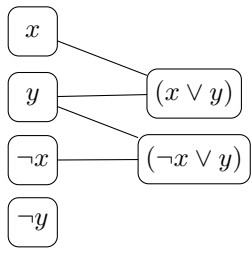

Figure 1: A graph representation for the formula $(x \lor y) \land (\neg x \lor y)$.

It may seem reasonable to treat Boolean formulas as text and apply embedding techniques such as word2vec (Mikolov et al., 2013). However, unlike words in natural language, individual variables in Boolean formulas are completely devoid of meaning. The meaning of variable $x$ in one formula is basically independent from variable $x$ in a second formula. This is amplified in our setting as we start from the QDI-MACS format, which does not admit naming variables with anything but a number. (See Appendix C for an example file.) Also sequence-based or tree-based LSTMs are hardly applicable: They would have to remember the possibly thousands of variable names and their associated semantics in the current formula in order to understand the interactions between distant occurrences of the same variable. In other words, the semantics of Boolean formulas in CNF arises from their structure, which we can represent as a graph as depicted in Fig. 1. The nodes in this graph are the literals and the clauses of the formula, and each clause has edges to the literals it contains.

We suggest the use of Graph Neural Networks (GNNs) to compute an *embedding* for every literal in the formula. Based on this embedding, we then use a policy network to predict the *quality* of each literal independently. Fig. 2 depicts an overview of our neural network architecture.

In this way, we only need to learn two operators: (1) to compute a clause embedding based on the embeddings of its literals, and (2) to compute the literal embedding based on the clause embeddings in which it occurs. Since the network is completely agnostic of variable names, it is easy to apply it to arbitrarily large formulas and unseen formulas. Through the careful choice of operations in the GNN, we can enforce the invariance of the network under reordering the clauses of the formula and reordering literals within each clause. We describe the GNN in Subsection 4.1 and the policy network in Subsection 4.2.

### 4.1 A GNN FOR BOOLEAN FORMULAS

The GNN maps the formula and the algorithm state to an embedding for each literal. For the embedding, we will only consider the part of the solver state that is specific to the variables and clauses. The information that goes into the GNN consists of the formula and the solver state. The formula is represented by the connectivity of the graph and the quantifiers for each variable. The (formula-specific) algorithm state only indicates for each variable whether it currently has a value assigned, whether it was selected previously in the same search branch, and the VSIDS activity score. For each clause, the algorithm state states if the clause was original or derived by the conflict analysis. We can summarize all this information, except for the graph structure, in a label $\mathbf{lit}$ of dimension $\lambda_L = 8$ for each literal and a label $\mathbf{c} \in \{0,1\}$ of dimension $\lambda_C = 1$ for each clause. See Appendix B for details.

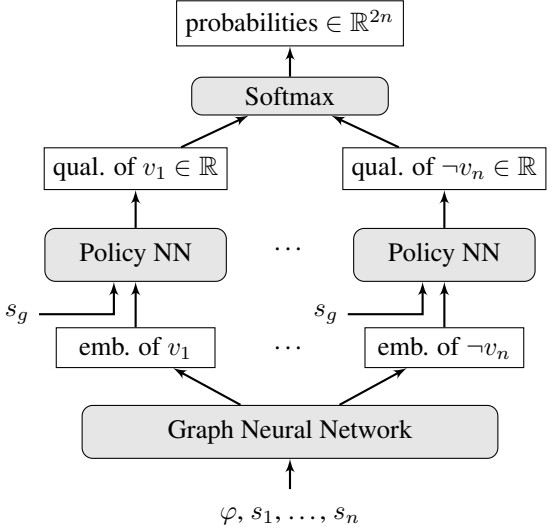

Figure 2: Computation graph for a formula $\varphi$ with $n$ variables, the global state $s_g$ of the solver, and the variable-specific state $s_1, \ldots, s_n$ of the solver.

Our *hyperparameters* include the embedding dimension $\delta_L \in \mathbb{N}$ for variables, the embedding dimension $\delta_C \in \mathbb{N}$ for clauses, and the number of iterations $\tau \in \mathbb{N}$ of the GNN. Trainable parameters of our model are indicated as bold capital letters. They consist of the matrix $\mathbf{W}_t^L$ of shape $(2\delta_L + \lambda_L, \delta_C)$, the vector $\mathbf{B}_t^L$ vector of dimension $\delta_C$, the matrix $\mathbf{W}_t^C$ of shape $(\delta_C + \lambda_C, \delta_L)$, and the vector $\mathbf{B}_t^C$ of dimension $\delta_L$.

We define the initial literal embedding to be $l_0 = \mathbf{0}$. For each $1 \le t \le \tau$, we define the literal embedding $l_t \in \mathbb{R}^{\delta_L}$ for every literal $l$ and the clause embedding $c_t \in \mathbb{R}^{\delta_C}$ for every clause $c \in C$ as follows:

$$c_t = \mathrm{ReLU}\left( \sum_{l \in c} \mathbf{W}_t^L [\mathbf{lit}^\top, l_{t-1}^\top, \bar{l}_{t-1}^\top] + \mathbf{B}_t^L \right), \qquad l_t = \mathrm{ReLU}\left( \sum_{c, l \in c} \mathbf{W}_t^C [\mathbf{c}^\top, c_t^\top] + \mathbf{B}_t^C \right)$$

**Sparse Matrices** The connectivity matrix of the formula graph can be huge (the largest of our graphs have several million nodes), but is typically very sparse. So in the implementation it is crucial to use sparse matrices to represent the adjacency information of the nodes, which is needed to compute the sums inside the definitions of $c_t$ and $l_t$.

### 4.2 POLICY NETWORK

The policy network predicts the quality of each literal based on the literal embedding and the global solver state. The *global solver state* is a collection of $\lambda_G = 30$ values that include the essential solver state as well as statistical information about the execution. We provide the details in Appendix A. The policy network thus maps the literal embedding $[\mathbf{lit}^\top, l_t^\top, \bar{l}_t^\top]$ concatenated with the global solver state to a single numerical value indicating the *quality* of the literal. The policy network thus has $\lambda_L + 2\delta_L + \lambda_G$ inputs, which are followed by two fully-connected layers. The two hidden layers use the ReLU nonlinearity. We turn the predictions of the policy network into action probabilities by a softmax (after masking the illegal actions).

The only information that flows from other variables to the policy network's judgement must go through the graph neural network. Since we experimented with graph neural networks with few iterations this means that *the quality of each literal is decided locally*. The rationale behind this design is that it is simple and computationally efficient.

# 5 EXPERIMENTS

We conducted several experiments to examine whether we can improve the heuristics of the logic solver CADET through our deep reinforcement learning approach. [2] The specific questions we try to answer are as follows:

Q1 Can we learn to predict good actions for a family of formulas?

Q2 How does the policy trained on short episodes generalize to long episodes?

Q3 Does the learned policy generalize to formulas from a different family of formulas?

Q4 Does the improvement in the policy outweigh the additional computational effort? That is, can CADET actually solve more formulas in less time with the learned policy?

## 5.1 DATA

Finding suitable data for our setting is not trivial. Different sets of formulas show very different statistical properties. For example, it is known that while CDCL solvers for SAT perform well on formulas from a broad range of applications, they are drastically outperformed by local search algorithms for randomly generated synthetic formulas (and vice versa). To draw any conclusions about the practical use of our approach, we want to learn heuristics for formulas that are related to applications, in particular form verification, synthesis, and planning. However, we found that many sets of formulas from interesting applications have one of three problems: (1) they contained few formulas that CADET could not solve already; (2) they contained only few formulas that CADET could solve at all in a reasonable time limit; (3) they contained only a handful of formulas in the first place. Each of these problems inhibits the demonstration of effective learning.

We settled for a set of formulas representing the search for reductions between collections of first-order formulas generated by Jordan & Kaiser (2013). This set, which we call *Reductions* in the following, contains formulas on any level of difficulty and is sufficiently large. It consists of 4500 formulas of varying sizes and with varying degrees of hardness. On average the formulas have 316 variables; the largest formulas in the set have over 1600 variables and 12000 clauses. We filtered out 2500 formulas that are solved without any heuristic decisions. In order to enable us to answer question 2 (see above), we further set aside a test set of 200 formulas, leaving us with a training set of 1835 formulas.

We additionally consider the 2QBF evaluation set of the annual competition of QBF solvers, QBFE-VAL (Pulina, 2016). This will help us to address question 3.

## 5.2 REWARDS AND TRAINING

We jointly train the encoder network and the policy network using REINFORCE (Williams, 1992). For each batch we sample a formula from the training set, and generate $b$ episodes by solving it multiple times. In each episode we run CADET for up to 250 steps using the latest policy. Then we assign rewards for each of these episodes and use them to estimate the gradient. We apply standard techniques to improve the training, including gradient clipping, normalization of rewards, and whitening of input data.

We assign a small negative reward of $-10^{-4}$ for each decision to encourage the heuristic to solve each formula in fewer steps. When a formula is solved successfully, we assign reward 1 to the last decision. In this way, we effectively treat unfinished episodes ($> 250$ steps) as if they take 10000 steps, punishing them strongly.

---

[2] We provide the code and data of our experiments at `https://github.com/<anonymized>`.

## 5.3 RESULTS

We trained the model described in Section 4 on the *Reductions* training set. We denote the resulting policy *Learned* and present the aggregate results in Figure 3 as a *cactus plot*, as usual for logic solvers. The cactus plot in Figure 3 indicates how the number of solved formulas grows for increasing decision limits on the *test set* of the *Reductions* formulas. In a cactus plot, we record one episode for each formula and each heuristic and determine the number of decisions. We then sort the runs of each heuristic by the number of decisions taken and plot the series. When comparing heuristics, lower lines (or lines reaching further to the right) are thus better, as they indicate that more formulas were solved in less time.

We see that for a decision limit of 200 (dashed line in Fig. 3), i.e. the decision limit during training, Learned solved significantly more formulas than either of the baselines. The advantage of Learned over VSIDS is about as large as VSIDS over purely random choices. This is remarkable for the field and we can answer Q1 positively.

The upper part of Fig. 3 shows us that Learned performs well far beyond the decision limit of 200 steps that was used during its training. Observing the vertical distance between the lines of Learned and VSIDS, we can see that the advantage of Learned over VSIDS even grows exponentially with an increasing decision limit. (Note that the axis indicating the number of decisions is log-scaled.) We can thus answer Q2 positively.

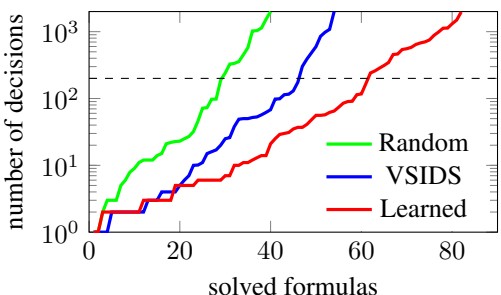

Figure 3: A cactus plot describing how many formulas were solved within growing decision limits. Lower and further to the right is better.

A surprising fact is that small and shallow neural networks already achieved the best results. Our best model uses $\tau = 1$, which means that for judging the quality of each variable, it only looks at the variable itself and the immediate neighbors (i.e. those variables it occurs together with in a constraint). The hyperparameters that resulted in the best model are $\delta_L = 16$, $\delta_C = 64$, and $\tau = 1$, leading to a model with merely 8353 parameters. The small size of our model was also helpful to achieve quick inference times.

To answer Q3, we evaluated the learned heuristic also on our second data set of formulas from the QBF solver competition QBFEVAL. Random solved 67 formulas, VSIDS solved 125 formulas, and Learned solved 101 formulas. So, while the policy trained on *Reductions* significantly improved over random choices, the generalization to this new formula set does not seem to be particularly strong, and it fell short of beating VSIDS.

Since our learning and inference implementation is written in python and not optimized, the performance cost of our approach is enormous compared to the pure C implementation of CADET using VSIDS. Comparing the blue dashed and solid lines in Fig 4, we can see that the original CADET (dashed line) is about 10x to 100x times faster than a version of VSIDS that goes through our python code, but does not even do inference (solid line). The learned heuristic must additionally do inference, slowing down its iterations even further. Despite its huge performance disadvantage, the learned heuristic reduces the overall execution time compared to the pure C implementation using VSIDS. The vertical gap between the curves towards the right end of the plot tells us that the advantage of the Learned heuristic can be as much as 90%. It also significantly improves in terms of the number of solved formulas within the time limit, which is the main performance criterion for logic solvers. Our last question Q4 can be answered positively.

## 6 RELATED WORK

Most previous approaches that applied neural networks to logical formulas used LSTMs or followed the syntax-tree (Bowman et al., 2014; Irving et al., 2016; Allamanis et al., 2016; Loos et al., 2017; Evans et al., 2018). Instead, we follow a GNN approach, where different occurrences of a variable are inherently connected, and the relevant information is propagated over multiple applications of a network. Additionally, this allows us to avoid fixed encodings of variables that are shared across

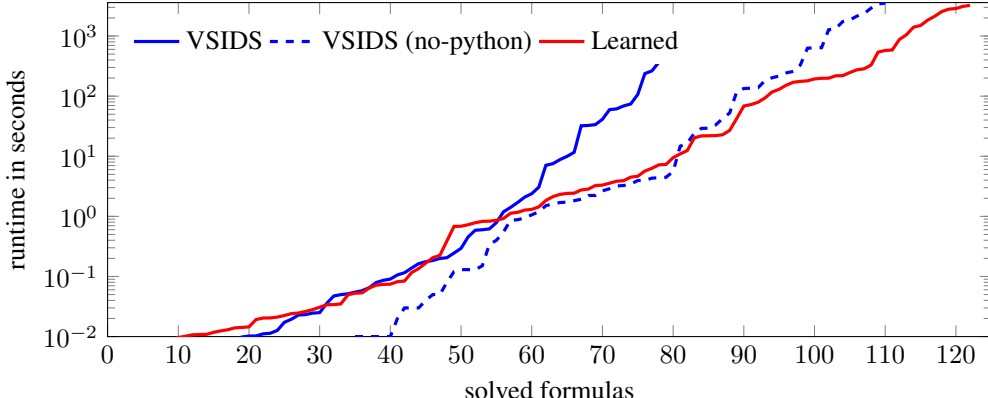

Figure 4: A cactus plot comparing the performance of different heuristics, measured in the wall clock time of CADET on formulas in the *Reductions* set. Lower and further to the right is better.

formulas. After all, variable $x$ in one formula has little in common with variable $x$ in another formula. Instead, GNNs allow us to compute embeddings of variables based on their context in the *current* formula.

Independent from our work, GNNS for formulas have been explored in NeuroSAT (Selsam et al., 2018). Also in their work, GNNs have shown far better scalability compared to previous approaches. While NeuroSAT relies solely on GNNs to predict the satisfiability of a given formula, we explore the combination of (graph) neural networks and logic solvers in a reinforcement learning setting. This allows us to leverage and improve the already impressive performance of logic solvers and scale to much larger formulas and have the guarantee that the answers produced are correct. (The algorithm we build on provides relatively succinct proofs for both True and False formulas.)

Dai et al. used GNNs to learn combinatorial algorithms over graphs (Dai et al., 2017). However, they focus on producing insights into the combination deep learning and combinatorial optimization and not on improving the state-of-the-art in the algorithmic problem they consider.

Previous learning approaches for SAT and MIP solvers focus on computationally cheap methods (e.g. SVMs) that learn within the run on a single formula (Liang et al., 2016; Khalil et al., 2016). Our focus is on learning across multiple formulas using (costly) deep reinforcement learning.

Other competitive QBF algorithms include expansion-based algorithms (Biere, 2004; Pigorsch & Scholl, 2010), CEGAR-based algorithms (Janota & Marques-Silva, 2011; 2015; Rabe & Tentrup, 2015), circuit-based algorithms (Klieber, 2012; Tentrup, 2016; Janota, 2018a;b), and hybrids (Janota et al., 2012; Tentrup, 2017). Recently, Janota successfully explored the use of (classical) machine learning techniques to address the generalization problem in QBF solvers (Janota, 2018a).

Reinforcement learning has been applied to other logic reasoning tasks. Kaliszyk et al. (2018) recently explored learning linear policies for tableaux-style theorem proving. Singh et al. (2018) learned heuristics for program analysis of numerical programs. Huang et al. (2018) created a learning environment for the theorem prover Coq. Kusumoto et al. (2018) applied reinforcement learning to propositional logic in a setting similar to ours; just that we employ the learning in existing strong solving algorithms, leading to much better scalability.

## 7 CONCLUSION

We presented an approach to improve the heuristics of a backtracking search algorithm for Boolean logic through deep reinforcement learning. The setting is new and challenging to reinforcement learning, featuring an unbounded input-size and action space, a connection between the length of episodes and rewards. We demonstrate that these problems can be overcome, and learn a heuristic that reduces the overall execution time of a competitive QBF solver by a factor of 10 after training on similar formulas.

We believe that this work motivates more aggressive research efforts in the area and will lead to a significant improvement in the performance of logic solvers. Our experiments uncover particu-

lar challenges to be addressed in the future. The transfer of the learned insights between different sets of formulas is still limited, learning from long episodes seems to be challenging, and—counterintuitively—it does not seem to help to consider multiple iterations of the GNN.

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

## A  GLOBAL SOLVER STATE

1. Current decision level
2. Maximal activity
3. Number of restarts
4. Restarts since last major restart
5. Conflicts until next restart
6. Number of variables
7. Ratio of universal variables to existential variables
8. Number of active clauses
9. Ratio of variables that already have a Skolem function to total variables
10. Number of decisions processed
11. Number of conflicts processed
12. Ratio of decisions and conflicts
13. Ratio of decisions and restarts
14. Number of successful propagation steps
15. Number of constants that were propagated
16. Ratio of constant propagations to function propagations
17. Number of pure variable assignments ($\approx$ decisions that cannot cause conflicts and are taken automatically)
18. Number of propagation checks (including unsuccessful checks)
19. Ratio of propagation checks that were successful
20. Number of local conflict checks
21. Number of global conflict checks
22. Ratio of local conflict checks that were successful
23. Ratio of global conflict checks that were successful (i.e. led to conflicts)
24. Number of conflicts caused by propagation of constants
25. Ratio of conflicts that were caused by constant propagation
26. Total length of learned clauses
27. Average length of learned clauses
28. Number of literals removed through clause minimization
29. Ratio of literals removed through clause minimization to total length of learned clauses
30. Maximum activity value over all variables

## B  VARIABLE LABELS

$y_0 \in \{0, 1\}$  indicates whether the variable is universally quantified,
$y_1 \in \{0, 1\}$  indicates whether the variable is existentially quantified,
$y_2 \in \{0, 1\}$  indicates whether the variable has a Skolem function already,
$y_3 \in \{0, 1\}$  indicates whether the variable was assigned constant True,
$y_4 \in \{0, 1\}$  indicates whether the variable was assigned constant False,
$y_5 \in \{0, 1\}$  indicates whether the variable was decided positive,
$y_6 \in \{0, 1\}$  indicates whether the variable was decided negative, and
$y_7 \in \mathbb{R}_{\geq 0}$  indicates the activity level of the variable.

## C  THE QDIMACS FILE FORMAT

QDIMACS is the standard representation of quantified Boolean formulas in prenex CNF. It consists of a header "`p cnf <num_variables> <num_clauses>`" describing the number of variables and the number of clauses in the formula. The lines following the header indicate the quantifiers. Lines starting with 'a' introduce universally quantified variables and lines starting with 'e' introduce existentially quantified variables. All lines except the header are terminated with 0; hence there cannot be a variable named 0. Every line after the quantifiers describes a single clause (i.e. a disjunction over variables and negated variables). Variables are indicated simply by an index; negated variables are indicated by a negative index. Below give the QDIMACS representation of the formula $\forall x. \exists y. (x \lor y) \land (\neg x \lor y)$:

```
p cnf 2 2
a 1 0
e 2 0
1 2 0
−1 2 0
```

There is no way to assign variables strings as names. The reasoning behind this decision is that this format is only meant to be used for the computational backend.

## D  HYPERPARAMETERS AND TRAINING DETAILS

We trained a model on the reduction problems training set for 10M steps on an AWS server of type C5. We trained with the following hyperparameters, yet we note that training does not seem overly sensitive:

- Literal embedding dimension: $\delta_L = 16$
- Clause embedding dimension: $\delta_C = 64$
- Learning rate: 0.0006 for the first 2m steps, then 0.0001
- Discount factor: $\gamma = 0.99$
- Gradient clipping: 2
- Number of iterations (size of graph convolution): 1
- Minimal number of timesteps per batch: 1200

## E  ADDITIONAL DATASETS AND EXPERIMENTS

While the set of Reductions-formulas we considered in the main part of the paper was created independently from this paper and is therefore unlikely to be biased towards our approach, one may ask if it is just a coincidence that our approach was able to learn a good heuristic for that particular set of formulas. In this appendix we consider two additional sets of formulas that we call *Boolean* and *Words*, and replicated the results from the main part. We show that we can learn a heuristic for a given set/distribution of formulas that outperforms VSIDS by a significant margin.

*Boolean* is a set of formulas of random circuits. Starting from a fixed number (8) of Boolean inputs to the circuit, individual AND-gates are added (with randomly chosen inputs with random polarity) up to a certain randomized limit. This circuit is turned into a propositional Boolean formula using the Tseitin transformation, and then a small fraction of random clauses is added to add some irregularities to the circuit. (Up to this point, the process is performed by the fuzz-tester for SAT solvers, FuzzSAT, available here `http://fmv.jku.at/fuzzsat/`.) To turn this kind of propositional formulas into QBFs, we randomly selected 4 variables to be universally quantified. This resulted in a more or less even split of true and false formulas. The formulas have 50.7 variables on average. In Figure 5 we see that training a model on these formulas (we call this model *Boolean*, like the data set) results in significantly better performance than VSIDS. The advantage of the learned heuristic over VSIDS and Random is smaller compared to the experiments on Reductions in the main part

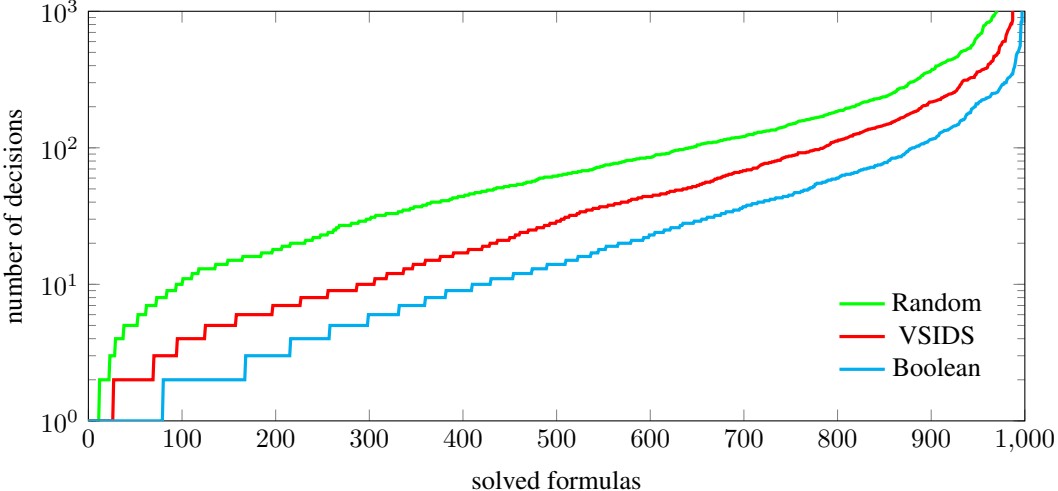

Figure 5: A cactus plot describing how many formulas were solved within growing decision limits on the *Boolean* test set. Lower and further to the right is better.

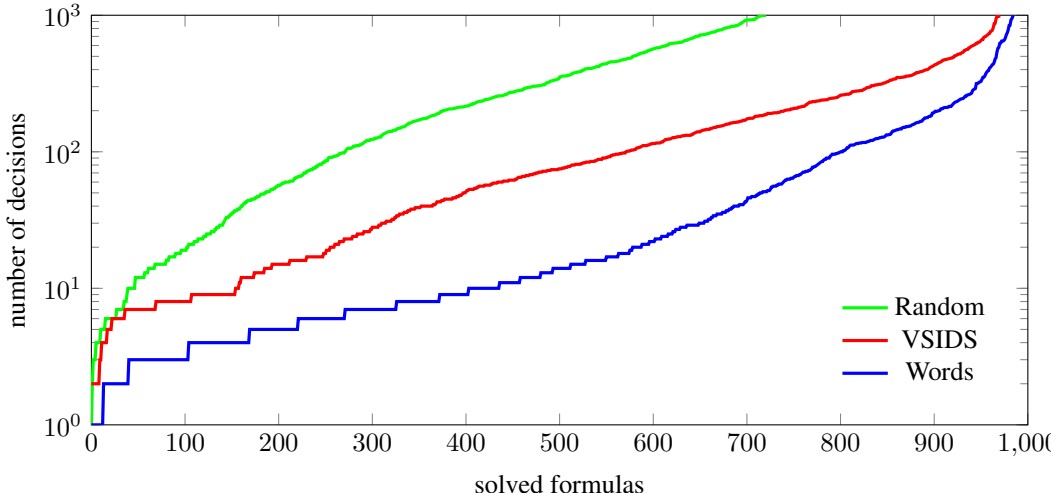

Figure 6: A cactus plot describing how many formulas were solved within growing decision limits on the *Words* test set. Lower and further to the right is better.

of the paper. We conjecture that this is due to the fact that these formulas are much easier to begin with, which means that there is not as much potential for improvement.

*Words* is a data set of random expressions over (signed) bitvectors. The top-level operator is a comparison ($=, \leq, \geq, <, >$), and the two subexpressions of the comparison are arithmetic expressions. The number of operators and leafs in each expression is 9, and all bitvectors have word size 8. The expressions contain up to four bitvector variables, alternately assigned to be existentially and universally quantified. The formulas are simplified using the circuit synthesis tool ABC, and then they are turned into CNF using the standard Tseitin transformation. The resulting formulas have 71.4 variables on average and are significantly harder for both Random and VSIDS. For example, the first formula from the data set looks as follows: $\forall z.\exists x.((x - z) \text{ xor } z) \neq z + 1$, which results in a QBF with 115 variables and 298 clauses. This statement happens to be true and is solved with just 9 decisions using the VSIDS heuristic. In Figure 6 we see that training a new model on the *Words* dataset again results in significantly improved performance. (We named the model *Words*, after the data set.)

We did not include the formula sets Boolean and Words in the main part, as they are generated by a random process - in contrast to Reductions, which is generated with a concrete application in mind.

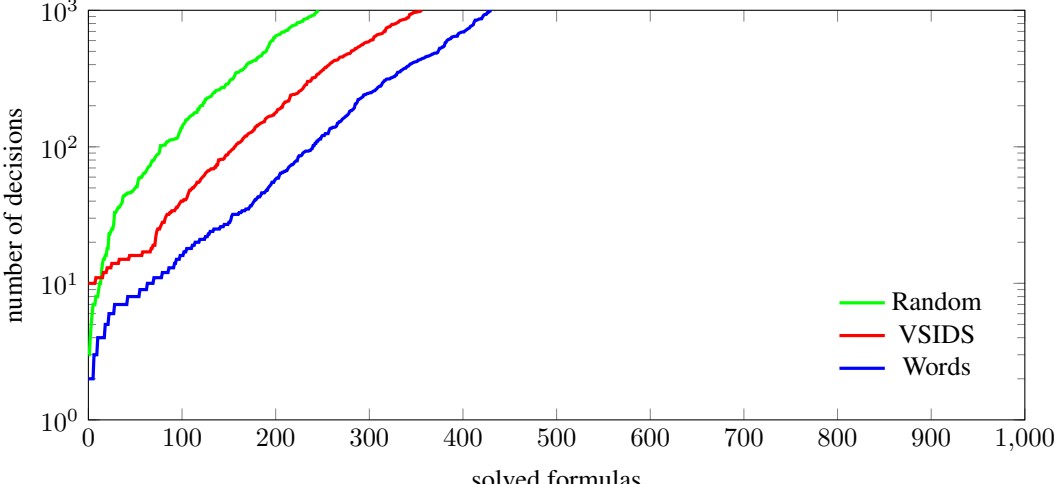

Figure 7: A cactus plot describing how many formulas were solved within growing decision limits on the *Words30* test set. Lower and further to the right is better. Note that unlike in the other plots, the model *Words* was not trained on this distribution of formulas, but on the same *Words* dataset as before.

In the formal methods community, artificially generated sets of formulas are known to differ from application formulas in non-obvious ways.

## F  ADDITIONAL EXPERIMENTS ON GENERALIZATION TO LARGER FORMULAS

An interesting observation that we made is that models trained on sets of small formulas generalize well to larger formulas from similar distributions. To demonstrate this, we generated a set of larger formulas, similar to the *Words* dataset. We call the new dataset *Words30*, and the only difference to *Words* is that the expressions have size 30. The resulting formulas have 186.6 variables on average. This time, instead of training a new model, we test the model trained on *Words* (from Figure 6) on this new dataset.

In Figure 7, we see that the overall hardness (measured in the number of decisions needed to solve the formulas) has increased a lot, but the relative performance of the heuristics is still very similar. This shows that the heuristic learned on small formulas generalizes relatively well to much larger/harder formulas.

In Figure 3, we have already observed that the heuristic also generalizes well to much longer episodes than those it was trained on. We believe that this is due to the "locality" of the decisions we force the network to take: The graph neural network approach uses just one iteration, such that we force the heuristics to take very local decisions. Not being able to optimize globally, the heuristics have to learn local features that are helpful to solve a problem sooner rather than later. It seems plausible that this behavior generalizes well to larger formulas (Figure 7) or much longer episodes (Figure 3).

