# OpenReview forum: "Learning Heuristics for Automated Reasoning through Reinforcement Learning"
_ICLR.cc/2019/Conference_

### Official Review · AnonReviewer1 · 2018-11-01
**Interesting application of deep learning with interesting results.**

**Rating:** 7
**Confidence:** 4

**Review:**

The paper proposes an approach to automatically learning variable selection
heuristics for QBF using deep learning. The evaluation presented by the authors
shows the promise of the method and demonstrates significant performance
improvements over a variable selection heuristic that does not use machine
learning.

In practice, the overhead of the proposed method is likely to be a major
obstacle in its adoption. The authors note the difficulty of finding suitable
benchmarks and restrict the set of instances they use for evaluation to formulae
where the proposed method is likely to achieve improvements. This skews the
evaluation in favor of the proposed method; in particular, the 90% improvement
figure mentioned in the abstract is not representative of the general case.
Indeed, on another set of instances the proposed method falls significantly
short of the performance of a state-of-the-art heuristic that does not employ
learning.

A drawback of the paper is that there is no comparison to related work. I
realize that this is difficult to achieve because other approaches are in
related, but different areas and may be difficult to adapt for this case, but a
general comparison to the improvements other approaches achieve would be
helpful.

Nevertheless, the work is interesting and presents a new angle on using machine
learning to speed up combinatorial problem solving. While several issues hinder
practical adoption, this is likely to lead to interesting follow-up work that
will improve problem solving in practice.

The description of the method (Section 4.1) is short and not detailed enough to
reproduce the approach the authors are proposing. However, the code is
available.

In summary, I feel that the paper can be accepted.

---

> ### Author Response · Authors · 2018-11-14
> **Clarification about QBFEVAL and additional data sets**
>
> Thank you for the detailed comments.
>
> >>> The authors note the difficulty of finding suitable benchmarks and restrict the set of instances
> >>> they use for evaluation to formulae where the proposed method is likely to achieve improvements.
> >>> This skews the evaluation in favor of the proposed method; in particular, the 90% improvement
> >>> figure mentioned in the abstract is not representative of the general case. Indeed, on
> >>> another set of instances the proposed method falls significantly short of the performance of
> >>> a state-of-the-art heuristic that does not employ learning.
>
> Our claim is that training a model on several hundred formulas greatly improves the performance of the logic solver on formulas from the same distribution. In our paper, we only support this claim by experiments on the Reductions benchmark. But, in fact, we have confirmed the same results on several datasets of artificially synthesized formulas (encoding random bit-level and word-level circuits). These additional datasets can be found with the published code, and we will provide more details about them in the appendix.
>
> It is natural to ask how a model trained on one distribution performs on a different dataset. Since QBFEVAL is an important data set in the formal methods community, we used it to test the transferability of the heuristics with only partial success. (Training a model directly on QBFEVAL does not seem to be possible at the moment, because of the small size of the dataset, leading to overfitting.)
>
> Lastly, we want to point out that most other works on ML for formulas only consider sets of random formulas (in particular, formulas synthesized by the authors themselves). In comparison, the Reduction benchmark is a well-known data set from the literature and generated independently from our work. In this way, we believe that we avoid skewing the results in our favor and set a higher bar than related work.
>
> >>> A drawback of the paper is that there is no comparison to related work. I
> >>> realize that this is difficult to achieve because other approaches are in
> >>> related, but different areas and may be difficult to adapt for this case, but a
> >>> general comparison to the improvements other approaches achieve would be
> >>> helpful.
>
> We would love to learn about (and compare to) related work, but we are not aware of any we could meaningfully compare to. Could you point us to any works you are aware of?
>
> Compared to the typical improvements through progress in hand-written heuristics, the 1000x improvement in the number of steps needed is enormous.

---

> > ### Comment · AnonReviewer1 · 2018-11-14
> > **Related Work**
> >
> > The most relevant paper to the work you're proposing here is probably "Learning to Branch in Mixed Integer Programming", https://dl.acm.org/citation.cfm?id=3015920.

---

> > > ### Author Response · Authors · 2018-11-16
> > > **Thanks for the interesting pointer!**
> > >
> > > We were not aware of this work, and will discuss it in our related work section. There are several key differences compared to our work: Khalil et al. present an approach to learn to predict an existing heuristic called SB using SVMs, while we attempt to learn an entirely new heuristics using deep reinforcement learning. Further, they learn within a single run of the solver, while we learn from executions on a set of formulas. In some sense, the approaches are quite orthogonal, and not necessarily competing against each other. It is unclear to us, if there is a meaningful way to compare the methods experimentally.

---

> > ### Author Response · Authors · 2018-11-27
> > **Additional experiments**
> >
> > As promised, we added additional experiments to the appendix. Please find them in Appendix E and Appendix F.

---

### Official Review · AnonReviewer2 · 2018-11-02
**needs some improvement**

**Rating:** 6
**Confidence:** 4

**Review:**

The aim of this paper is to learn a heuristic for a backtracking search algorithm utilizing Reinforcement learning. The proposed model makes use of Graphical Neural Networks to produce literal and clauses embeddings, and use them to predict the quality of each literal, through a NN, which in turn decides the probability of each action.

Positives
A new approach on how to employ Machine learning techniques to Automated reasoning problems. Works with any 2QBF solver.
The learned heuristic seems to perform better than the state of the art in the presented experiments.

Negatives
No theoretical justification about why this heuristic should work better than the existing ones.
Doesn't solve QBF formulas in general, but only 2QBF.
It is not clear whether the range of formulas that can be solved using this approach is greater than that of existing solvers.
Having a substantial amount of formulas that produce incomplete episodes, as it might be the case in real world scenarios, hinders learning, so the dataset has to be manually adjusted.

Conclusion
The proposed framework is an interesting addition to existing techniques in the field and the idea is suitable for further exploration and refinement. The experimental results are promising, so the direction of the work is worth pursuing. However, some of the foundations and overall nature of the work needs some improvement and maturity.

---

> ### Author Response · Authors · 2018-11-14
> **Some remarks about the concerns raised**
>
> We thank the reviewer for the detailed feedback.
>
> >>> No theoretical justification about why this heuristic should work better than the existing ones.
>
> This is a very interesting question, but surprisingly hard to answer. Even for the simpler question of why CDCL for SAT solvers is so unreasonably effective for a wide range of applications, there is no concrete theoretical explanation - despite two decades of research! When there is no satisfactory theoretical explanation, we suggest that it is better to learn the heuristics based on the data itself.
>
> >>> Doesn't solve QBF formulas in general, but only 2QBF.
>
> Our approach could be easily applied to general QBF as well. The limitation to 2QBF is also due to the underlying tool. But keep in mind that most applications of QBF, e.g. in verification and program synthesis, can be encoded with just one quantifier alternation, so we believe that we captured the most interesting cases of QBF.
>
> >>> It is not clear whether the range of formulas that can be solved using this approach is
> >>> greater than that of existing solvers.
>
> Our experiments demonstrate that we can solve significantly more formulas when given enough formulas from a single source (=distribution). We do not claim that the learned models generalize to formulas far away from that distribution. The question whether it is possible to learn models that apply to a wide “range of formulas” is indeed an open one.
>
> >>> Having a substantial amount of formulas that produce incomplete episodes, as it might be
> >>> the case in real world scenarios, hinders learning, so the dataset has to be manually
> >>> adjusted.
>
> We believe that this the inherent challenge of problem solving: how can we learn to solve problems that we have never solved? The assumption underlying this paper is that learning how to solve simpler problems faster, helps us to solve harder problems, too. Our experiments demonstrate that this is indeed possible for problems sets containing many related formulas of different hardness levels.

---

### Official Review · AnonReviewer3 · 2018-11-02
**Interesting application of reinforcement learning and GNN over a specific decision problem**

**Rating:** 5
**Confidence:** 3

**Review:**

The paper is proposing to use reinforcement learning as a method for implementing heuristics of a backtracking search algorithm or Boolean Logic. While I'm not familiar with this specific topic, Section 2 is didactic and clear. The challenges of the tackle problem are clearly explained in this section.

The Graph neural network architecture proposed in Section 4 to compute literals of the formula is an original idea. The experimental results look convincing and suggest this approach should be more deeply investigated.

My main concern is that the novelty from a machine learning and reinforcement learning point of view remains limited while the application seems original and promising. So I will not be strongly opposed to the publication if this work in ICLR venue while I remain unsure it is the best one.

---

> ### Author Response · Authors · 2018-11-14
> **We believe the work contains insights for the ML community, too.**
>
> We thank the reviewer for the insightful comments.
>
> >>> [...] the novelty from a ML and RL point of view remains limited [...]
>
> We see contributions to two lines of work published in ICLR and related conferences: The first concerns the representation of formulas to facilitate learning [1, 2, 3], and the second concerns leveraging reinforcement learning in combinatorial search algorithms [5, 6].
>
> Compared to [1, 2, 3], we show how to address the problem of scale. Previous works suggested tree-encoders [2], possible worlds [1], and top-down tree encoders [3]. These approaches seem to be limited to formulas with tens of variables, which would be considered tiny in the verification/formal methods community. To scale up to realistic formulas, orders of magnitude larger of what has been considered before, we suggest to exploit the graph representation of formulas in conjunctive normal form and apply GNNs. While GNNs generally scale well, this is also a conceptual shift: Previous works needed to learn a fixed embedding for variables “a”, “b”, “c”, etc., even though variable “a” has no shared meaning across different formulas. GNNs enable us to embed variables based only on the context of their occurrences in the current formula.
>
> Compared to [5, 6], our work represents a big step towards practicality. While interesting from a learning perspective, their methods do not come even close to the state-of-the-art in specialized algorithms. We demonstrate that the tight integration of deep learning and combinatorial search algorithms can actually improve the performance of complex and (relatively) large-scale applications of combinatorial search. The main challenge here was the significant performance cost of neural networks. Our work shows that this cost can be outweighed by the dramatically better decisions neural networks suggest (1000x fewer steps needed to solve hard formulas).
>
> We acknowledge that we need to state these points more clearly, and will improve the paper accordingly.
>
> [1] "Can Neural Networks Understand Logical Entailment?", in ICLR 2018
> [2] "Learning Continuous Semantic Representations of Symbolic Expressions", in ICML 2017
> [3] "Top-down neural model for formulae", under submission to ICLR 2019
> [4] "Learning a SAT Solver from Single-Bit Supervision", under submission to ICLR 2019
> [5] "Learning Combinatorial Optimization Algorithms over Graphs", in NIPS 2017
> [6] "Neural Combinatorial Optimization with Reinforcement Learning", in ICLR 2017

---

### Meta-Review · Area_Chair1 · 2018-12-13
**Borderline paper**

**Confidence:** 3
**Recommendation:** Reject

**Metareview:**

The paper proposes the use of reinforcement learning to learn heuristics in backtracking search algorithm for quantified boolean formulas, using a neural network to learn a suitable representation of literals and clauses to predict actions. The writing and the description of the method and results are generally clear. The main novelty lies in finding a good architecture/representation of the input, and demonstrating the use of RL in a new domain. While there is no theoretical justification for why this heuristic should work better than existing ones, the experimental results look convincing, although they are somewhat limited and the improvements are dataset dependent. In practice, the overhead of the proposed method could be an issue. There was some disagreement among the reviewers as to whether the improvements and the results are significant enough for publication.